# Identification of the Profile of the Patients with Hemophilia B Eligible for Treatment with Nonacog Alfa Once-Weekly

**Dorina Cultrera [1], Raimondo De Cristofaro [2], Paola Giordano [3], Silvia Linari [4], Silvia Macchi [5], Renato Marino [6], Angelo Claudio Molinari [7], Angiola Rocino [8], Cristina Santoro [9], Piercarla Schinco [10], Sergio Siragusa [11], Giuseppe Tagariello [12], Annarita Tagliaferri [13], Ezio Zanon [14] and Massimo Morfini [15,*]**

1   General Paediatric Dept., Haemophilia Centre, G. Garibaldi University Hospital, 95100 Catania, Italy; doricu@tiscali.it
2   Cuore, Policlinico Gemelli, Università Cattolica S, 00100 Rome, Italy; rdecristofaro@rm.unicatt.it
3   Paediatric Department, Giovanni XXIII University Hospital, 70121 Bari, Italy; paola.giordano@uniba.it
4   General Paediatric Dept., Haemophilia and Thrombosis Center, AOU Careggi University Hospital, 50100 Florence, Italy; linaris@aou-careggi.toscana.it
5   Haemophilia Centre, Ospedale Santa Maria delle croci Regional Hospital, 48121 Ravenna, Italy; silvia.macchi@auslromagna.it
6   Haemophilia Centre, AOU Bari University Hospital, 70121 Bari, Italy; renato.marino@policlinico.ba.it
7   Regional Reference Centre for Haemorrhagic Diseases, "Giannina Gaslini" Institute, 16121 Genoa, Italy; aclaudiomolinari@gaslini.org
8   Haemophilia Centre, San Giovanni Bosco Regional Hospital, 80100 Naples, Italy; angiolar@tin.it
9   Haemophilia Centre, La sapienza University Hospital, 00100 Rome, Italy; santoro@bce.uniroma1.it
10  Haematology Dept, Regional Hospital "Molinette", 10121 Turin, Italy; pcschinco@hotmail.com
11  Haematology Department, University Hospital of Palermo, 90121 Palermo, Italy; sergio.siragusa@unipa.it
12  Haemophilia Center and Blood Transusion Dept, Regional Hospital, 31033 Castelfranco Veneto, Italy; Giuseppe.tagariello@gmail.com
13  Haemophilia Centre, AOU Parma University Hospital, 43121 Parma, Italy; atagliaferri@ao.pr.it
14  Haemophilia Centre, AOU Padua University Hospital, 35100 Padua, Italy; ezio.zanon@unipd.it
15  Italian Association of Haemophilia Centres (AICE), 50100 Florence, Italy
*   Correspondence: drmassimo.morfini@gmail.it

**Abstract:** This study aimed to identify the characteristics of patients with hemophilia B eligible for once-weekly treatment with Nonacog alfa. Methods: A survey was conducted in 14 Hemophilia (HCs) of Italy. These centers were given a questionnaire consisting of ten closed multiple-choice questions. The centers were asked: (a) the percentages of their hemophilia B (HB) patients undergoing replacement therapy, "On-demand", or weekly prophylaxis, (b) the criteria guiding the monitoring of patients, the advantages according to the age of patients, and (c) the obstacles to prophylaxis. The percentage of patients receiving "On-demand" (OD) treatment or continuous prophylaxis (prophy) differed depending on patient age and the severity of the disease. Only 57% of HCs provided "On-demand" therapy to the mild HB patients, about 93% to moderate ones, of whom 43% on prophylaxis. About 78% of patients <6 years old, were on treatment in 9 out of 14 HCs, by prophylaxis 66.7% and 33.3% by On-demand. In the 6–18 age group, 90.1% of HCs treated HB patients with prophylaxis, 42.8% in the 18–30 age range. On-demand treatment was the therapy of choice in 61.5% of HCs for patients aged 30–65 years. In total, 64% of the HCs assigned the maximum score to bleeding frequency, especially in the <6 and 6–18 age groups. Bleeding severity was also taken into significant consideration, particularly in subjects up to 30 years old. The scores regarding venous access were distributed relatively evenly throughout all age groups. The majority of the centers attributed a medium-high score to treatment compliance, especially in the 6–65 age range. In actuality, 55% of HCs attributed pro-thrombotic comorbidity a low score in the 18–30 age

group, whereas 81% gave pro-hemorrhagic comorbidity a high rating in patients aged >65 years old. Many centers assigned a medium-high score to the baseline concentration of FIX level at diagnosis in all age groups. Most HCs attributed a medium-high score to type of genetic mutation in the younger age groups. As for socio-cultural barriers and quality of life, the majority of respondents gave a medium-high score in all age groups. For periodic monitoring of patients receiving continuous prophylaxis, 59% of the centers reported using clinical assessment. With regard to prophylaxis administration method, the majority of hemophiliacs were given infusions twice weekly, while as regards to the dose of FIX concentrate delivered, 50% of the centers reported administering prophylaxis once-weekly at a dose ranging from 5–100 IU/kg in 10–50% of HB patients. Thus, 93% of the centers reported using a dose of 25–50 IU/kg for twice-weekly prophylaxis in 6–100% of the patients. The majority of centers (86%) believe that, in a program of early primary prevention, once-weekly treatment with nonacog alfa may represent an alternative strategy to dose escalation. The results show that patients with mild hemophilia, with functional musculoskeletal status and difficulties with venous access, are candidates for once-weekly prophylaxis with nonacog alfa. For such patients, this regimen can improve treatment compliance and quality of life.

**Keywords:** hemophilia B; factor IX; nonacog alfa; once-weekly prophylaxis

## 1. Introduction

Hemophilia B is an X-linked coagulation disorder characterized by a deficiency of functionally active coagulation factor IX (FIX), resulting in spontaneous or trauma-induced bleeding primarily in joints, muscles, and soft tissues. Prevention and treatment of bleeding episodes in patients with hemophilia B are based on prophylactic or "On-demand" replacement of the deficient FIX with plasma-derived or recombinant FIX (rFIX) products (including Fc fusion proteins) [1–3].

Outstanding improvement in FIX half-life has recently been achieved through pegylation or albumin fusion of the rFIX molecule, creating what is known as rFIX Extensive Half-life (rFIX EHL) concentrates [4,5], substances which allow very long intervals (14–21 days) between prophylactic infusions. Despite the improved half-life of the new rFIX EHL concentrates, their higher cost has caused a significant increase in hemophilia B treatment in the USA [6], challenging the cost/effectiveness ratio of this new therapy. For this reason, a survey among clinicians treating hemophilia B was conducted in Italy to evaluate the feasibility and safety of once-weekly prophylaxis with Nonacog alfa. Prophylaxis with FIX products is endorsed by the World Health Organization (WHO) [7], the World Federation of Hemophilia (WFH) [8], and the National Hemophilia Foundation (NHF) [9] as the primary therapy for individuals with severe hemophilia B. Primary prophylaxis should be initiated at three years of age in the absence of documented osteochondral joint disease (determined by physical examination and/or imaging studies), before the second clinically evident significant joint bleed; secondary prophylaxis should be started following two or more bleeds into large joints, before the onset of documented joint disease [3]. However, only 21–55% of hemophilia B patients receive prophylactic therapy [10–12]. Underuse of prophylaxis is more evident in hemophilia B patients than in hemophilia A patients: one survey showed that the frequency of prophylaxis for severe hemophilia A was more than twice that for severe hemophilia B (69% vs. 32%), with the discrepancy being most marked in patients between 18 and 30 years of age (70% vs. 20%) [13]. The benefits of prophylaxis, in terms of preventing joint bleeding and its associated deleterious effects on joint function and structure (hemophilic arthropathy), have been reported in patients with hemophilia B. However, much of the data stems from small subpopulations within more extensive clinical trials [14–17]. With the paucity of published clinical trials evaluating prophylactic dosing vs. "On-demand" regimens in hemophilia B populations, evidence-based conclusions regarding the benefits of prophylaxis with FIX products, while intuitively positive, remain an open question [18,19].

Nonacog alfa, the first rFIX product available for hemophilia B, is obtained from genetically engineered Chinese hamster ovary cells, with no exposure to animal or human plasma-derived components during its manufacture or formulation [20].

Nonacog alfa was licensed for the control and prevention of hemorrhagic episodes and routine and surgical prophylaxis in hemophilia B patients in 1997 in the USA, and 1998 in Europe. In 2007, Nonacog alfa and octocog alfa were reformulated with a change in the diluent to minimize the risk of red cell agglutination in tubing or syringes [21,22].

A phase 4 exploratory, open-label, randomized, 56-week study compared the efficacy and safety of two secondary prophylactic regimens of Nonacog alfa (50 IU/kg twice weekly or 100 IU/kg once-weekly) with "On-demand" treatment in patients with moderately severe to severe hemophilia B [23]. Both prophylactic regimens significantly reduced the annualized bleeding rate (ABR) compared to "On-demand" treatment, with no significant differences in ABR between the two prophylactic regimens. Once-weekly prophylaxis appeared to be an attractive therapeutic approach for patients with hemophilia B, although further evaluation was needed to confirm the safety and efficacy data reported in the study [23].

The efficacy and safety of 12 months of prophylaxis with once-weekly Nonacog alfa 100 IU/kg were subsequently compared to six months of "On-demand" treatment in adolescent and adult patients with moderately severe to severe hemophilia B [24]. Once-weekly prophylaxis with 100 IU/kg was associated with a lower ABR compared to that observed during "On-demand" treatment. The once-weekly prophylaxis was well tolerated, with a similar safety profile to that reported during the "On-demand" treatment period [24].

This survey was conducted using a questionnaire aimed to define the ideal profile of patients with hemophilia B to be proposed as candidates for once-weekly prophylaxis with Nonacog alfa [25]. Since chronic conditions such as hemophilia B require long-term treatment, a significant challenge of prophylactic regimens is adherence, particularly in adolescents and adults [26]. Several factors may contribute to this, including the failure of patients to see the benefits of prophylaxis, especially if they are asymptomatic, and the inconvenience and frequency of administration [27,28]. Once-weekly prophylaxis may, therefore, be a viable treatment option for adolescents and adults with hemophilia B [24].

## 2. Methods

The survey involved 14 Italian Hemophilia Treatment Centers (HTC), selected from the 43 Directors of the Italian Association of Hemophilia Centers (AICE). AICE members manage congenital coagulation diseases and are equally distributed throughout Italy. It was decided to include only AICE centers in the survey, since these HTCs ensure high clinical and organizational standards, and manage almost all hemophilia B patients in Italy. The AICE centers were given a questionnaire consisting of ten closed multiple-choice questions, delivered by e-mail. The HTCs were asked:

1. The percentages of their hemophilia B patients undergoing replacement therapy, "On-demand" treatment or continuous prophylaxis, depending on the severity of the disease;
2. The percentage distribution of patients, by age, receiving the three treatment above options;
3. The criteria guiding the choice to use continuous prophylactic treatment instead of "On-demand" therapy by age group;
4. The criteria used to monitor patients on continuous prophylaxis;
5. The percentage distribution of patients on once-weekly and twice-weekly prophylaxis and the doses of FIX used;
6. The role of Nonacog alfa in early primary prophylaxis;
7. The advantages of a once-weekly infusion scheme in patients on continuous prophylaxis;
8. The possible obstacles to once-weekly infusion prophylaxis in patients with difficult venous access;
9. The age group of patients they believe could benefit most from once-weekly prophylactic treatment,
10. To define the profile of patients eligible for once-weekly prophylaxis.

## 3. Results

### 3.1. Use of Replacement Therapy, "On-demand" Treatment or Continuous Prophylaxis Based on Disease Severity

Mild hemophilia B: 61.5% (8/13) of HTCs had the policy of providing On-demand therapy to 90–100% of their patients, while 38.5% (5/13) only administered this type of treatment to 40–57% of patients. Only one Center reported having some patients on prophylaxis.

Moderate hemophilia B: in 92% (12/13) of the HTCs, 80–100% of the patients were receiving replacement therapy, in 7 HTCs only by OD, and in 6 HTCs by both On-demand or Prophylaxis; the ratio of Prophy/OD was 0.70.

Severe hemophilia B: almost all the 14 HCs reported treating 90–100% of HB patients, 21.5% (3/14) exclusively with prophylaxis. The other 11 HCs were treating their HB patients with both OD and Prophy: The Prophy/OD ratio was 3.02. The answers of medical doctors participating to the Advisory Board are reported in Table 1

**Table 1.** Indications to treatment of the Directors of Hemophilia Centers according to the severity of the disease.

| | **Answers of Hemophilia Treatment Center (HTC) Directors** | | | | | |
|---|---|---|---|---|---|---|
| | **Patients on Therapy** | **HTC** | **Only On-Demand** | **Only Prophylaxis** | **Prophylaxis or On-Demand** | |
| **Severity Levels of Hemophilia** | **%** | *n* | *n* | *n* | *n* | **Patients' Ratio Prophy/OD (Mean of %)** |
| Mild: FIX 5–40 IU/dL | 90–100 | 8/13 | 12 | 0 | 1 | 0.24 |
| | 40–57 | 5/13 | | | | |
| Moderate: FIX 1–5 IU/dL | 80–100 | 12/13 | 7 | 0 | 6 | 0.70 |
| | 60 | 1/13 | | | | |
| Severe: <1 IU/dL | 90–100 | 14/14 | 0 | 3 | 11 | 3.02 |

### 3.2. Use and Type of Replacement Therapy According to Age

Only 9 HCs had on charge HB patients <6 years old, seven were treating all their patients, two only 10%. Only two HTC were treating their patients exclusively by continuous prophylaxis, seven by both Prophylaxis or On-Demand: the ratio Prophy/OD was 4.19.

As regards the 6-18-year-old age group, only 11 out of 14 (78.5%) HTCs had on charge patients in this age range at the time of the survey: 6 were treating 90–100% and five about 45–75% of their patients. Two or three HTCs were using exclusivelyOn-demand or Prophylaxis, respectively, and 6 HTCs were treating their patients by both Prophylaxis or On-Demand: the ratio Prophy/OD was 2.46. Concerning the group of patients 18–30 years old, 7 out of 11 HTCs were treating 90–100% of their patients, 5 only 25–80%. Two and one HTCs declared to treat their patients exclusively by OD or Prophy, respectively. The majority of HTCs, 9 out of 11 (82%) were using both Prophylaxis or OD: the ratio Prohy/OD was 1.21

In the 30–65 years of the age range, 9 and 4 HTCs out of 13 (69.2% and 30.8%, respectively) reported treating 90–100% or 20–80% of their HB patients respectively, two HTCs exclusively using prophy, none used OD. In total, 11 HTCs out of 13 (84.6%) administered the therapy either by OD or prophy: the ratio Prophy/OD was 1.05.

Overall, 9 HTCs and 4 out of 13 (69.2% and 30.8%, respectively) reported treating 90–100% and 5–65% patients in the >65 age group, respectively. Eight HTCs were using only On-Demand therapy and only one continuous Prophylaxis. The ratio Prophylaxis/OD was 0.35. Indications to treatment are reported in the following Table 2.

**Table 2.** Indications to treatment of the Directors of Hemophilia Centers according to the patients' age.

| | Answers of HTC Directors | | | | | |
| | Patients on Therapy | HTC | Only On-Demand | Only Prophylaxis | Prophylaxis or On-Demand | |
| Severity Levels of Hemophilia | % | *n* | *n* | *n* | *n* | Patients' Ratio Prophy/OD (Mean of %) |
|---|---|---|---|---|---|---|
| <6 years old | 100 | 7/9 | 0 | 2 | 7 | 4.19 |
| | 10 | 2/9 | | | | |
| 6–18 years old | 90–100 | 6/11 | 2 | 3 | 6 | 2.46 |
| | 45–75 | 5/11 | | | | |
| 18–30 years old | 90–100 | 7/11 | 2 | 1 | 9 | 1.21 |
| | 20–85 | 5/11 | | | | |
| 30–65 years old | 90–100 | 9/13 | 0 | 2 | 11 | 1.05 |
| | 15–80 | 4/13 | | | | |
| >65 years old | 90–100 | 9/13 | 8 | 1 | 4 | 0.35 |
| | 5–65 | 4/13 | | | | |

*3.3. The Criteria Guiding the Choice of Continuous Prophylaxis Rather Than "On-Demand" Treatment, by Age Group*

Ten out of 11 (90.1%) HCs assigned the highest scores (9 or 10, from a range 1–10) to the bleeding rate for all HB patients, but only 8 HCs (72.7%) scored 9 or 10 in patients >65 years. Bleeding severity scored 9 or 10 for all HB patients in 8/11 HCs (72.7%), except for patients >65 years old who received the same scores from 6/11 (54.5%) HCs. No definite opinion emerged regarding venous access, as the answers were spread out over several scores, high and low. Compliance with treatment scored eight in HB patients aged 6–18 years in 5/11 HCs (45.5%), an element that received the same score in patients <6 and 30–65 years old from 4/11 (36.4%) and 4/14 (28.5%) of HCs respectively. Pro-thrombotic comorbidities scored low values: a score of 1, according to 6/11 (54.5%) of HCs, was given for patients <6–18 years old, while a score of seven was assigned to patients 30–65 years old by 4/14 (28.5%) HCs. Pro-hemorrhagic comorbidities received a high score of 8 for patients 30–65 and >65 years old, in 4/14 (28.5%) and 5/11 (45.4%) of HCs, respectively.

Laboratory parameters used to decide between continuous prophylaxis and "On-demand" therapy were analyzed according to age group, the baseline level of FIX at diagnosis scored a high level of 10 in all age ranges, according to 3/11 (27.2%) HCs. Forty percent of the HTCs attributed the score of 5 out of 10 to genetic mutation type in <6-year age group, while for the >65-year age group, 40% of the HTCs awarded the score of 1 out of 10 to this item.

Socio-cultural barriers received a score of 8 out of 10 in the <6-year age group: 45% of the HTCs attributed 18% of them the score of 4 out of 10, while 9% each of the HTCs gave scores of 1, 2, 3 and 7 out of 10. For the 6–18 year age group, 5/11 (45%) of the HTCs attributed these barriers a score of 8 out of 10, while 3/14 (21.4%) and 3/11 (27.3%) of HCs scored the same value for HB patients aged 30–65 and >65 years old, respectively.

The importance of quality of life when deciding which type of treatment to adopt was awarded a score of 7 out of 10 in the <6 and 6–18 years age groups by 4/11 HCs (36%), while 3 (27%) of HTCs gave a score of 8 out of 10. For the age group 18–30 years, 4/13 (31%) of the HTCs attributed scores of 7 out of 10 and 8 out of 10 to this item. For the age group 30–65 years, 6/14 (43%) of the HTCs awarded this factor a score of 7 out of 10. For the age group >65 years, 4/11 (36%) of the HTCs attributed to the quality of life a score of 5 out of 10.

*3.4. The Criteria Used for Monitoring Patients on Continuous Prophylaxis*

Regarding the criteria adopted by the reference HTCs involved in this survey for the treatment of hemophilia, used to carry out regular monitoring of patients on continuous prophylaxis, it was found that 59% of the HTCs use clinical criteria. In contrast, 32% and 9% of the HTCs reported using the pre-infusion residual level of FIX and the pharmacokinetics of the infused FIX, respectively.

*3.5. Percentage Distribution of Patients on Once and Twice-A-Week Prophylaxis and Dosage Used*

Only 7/14 (50%) HTCs reported treating 10–50% of their patients with once-weekly prophylaxis, at a dosage ranging from 50–100 or 90–100 IU/kg. All HTCs recommended twice-weekly prophylaxis to their 80–100% patients at dose 25–50 IU/kg or to 10–50% of patients at dosage 25–40 IU/Kg. Recommandations for prophylaxis are reported inn Table 3.

**Table 3.** Prophylaxis regimens adopted by the Directors of the Hemophilia Centers.

| Prophylaxis | Answers of HTC | | Patients Treated | Dose |
| --- | --- | --- | --- | --- |
| | *n* | % | % | IU/kg |
| Once a week | 4/7 | 57 | 20–50 | 50–100 |
| | 3/7 | 43 | 10–15 | 90–100 |
| Twice a week | 11/14 | 78 | 80–100 | 25–50 |
| | 3/14 | 22 | 10–50 | 25–40 |

*3.6. The Role of Nonacog Alfa in Weekly Prophylaxis in HB*

With regards to the role of Nonacog alfa in early primary prophylaxis, 86% of the HTCs believed that once-weekly treatment might represent an alternative strategy to dose escalation. In contrast, the remaining 14% opposed this regimen due to the half-life of this concentrate, considering twice-weekly prophylactic therapy more appropriate.

*3.7. Possible Advantages of Once-Weekly Infusions in Patients on Continuous Prophylaxis.*

As regards the advantages of once-weekly therapy in patients on continuous prophylaxis, the distribution of the responses from the HTCs was heterogeneous.

The relevance of savings in total expenditure was attributed to a score of 2 and 4 out of 5 by 4/11 (31%) of the HTCs. Regarding the advantage of better compliance, 5/13 (38.5%) of the HTCs attributed this factor a score of 3 out of 5, 23.1% of them the score of 5 out of 5. As for a lower risk of infection, 5/13 (38%) of the HTCs assigned scores of 1 out of 5, while the other 5 (38%) awarded a score of 5 out of 5.

The advantage of a better quality of life was assigned a score of 3 out of 5 by 5/13 (38%) of the HTCs, and a score of 1 out of 5 by 3/13 (23%) HTCs. Easier management of patients with difficult venous access was rated heterogeneously, with 4/13 (31%) of the HTCs assigning scores of 2 out of 5, and the other 4/13 5 out of 5.

*3.8. Possible Obstacles to Once-Weekly Infusion Prophylaxis in Patients with Difficult Venous Access*

The responses of the HTCs regarding perceived barriers to the use of a once-weekly prophylactic regimen in patients with difficult venous access were distributed heterogeneously. Regarding the limited pharmacokinetic data on the once-weekly regimen, 9 out of 14 (64.3%) HCs rated this item between 4 and 6. Regarding patients' refusal/inability to follow the prophylactic regimen, 6 out of 13 (46.1%) HCs awarded this parameter a very low score (1–2), and 5 out of 13 (38.4%) awarded a score of 4–5. Technical difficulties with administering infusions regularly were scored 4–3 by seven out of 14 (50%) of HCs. Concerning a preference expressed by a patient, 7 out of 14 (50%) HCs awarded the score 3–4, and three out 14 (21.5%) a score of 6. The hemorrhagic phenotype was scored 4–6 by nine out of 14 (64.2%) of HCs. Concerning a risk of poor compliance, nine out of 14 (64.2%) HCs awarded scores of 1–3.

*3.9. The Age Group of Patients That Could Benefit Most from Once-Weekly Prophylactic Treatment*

Ten HCs out of 13 (76.9%) assigned a score of 4–5 to the age group <6, while eight out of 13 (61.5%) HCs awarded the same score to the 6–18 years patient group. Six HCs out of 13 (46.1%) awarded scores of 4–5 to the >65 year HB patient group. The reasons and advantages of prophylaxis according to the medical doctors participating to the advisory board are reported in Table 4.

**Table 4.** The answers and reasons according to the Directors of HTCs about the age group of patients that could benefit most from once-weekly prophylactic treatment.

| Patients' Group According to Age | SCORES | | | | | Reasons for Decision |
|---|---|---|---|---|---|---|
| | **1** | **2** | **3** | **4** | **5** | |
| | **Answers of HTCs** | | | | | |
| <6 years old | 2 | 1 | 0 | 3 | 7 | Better parents' compliance; Difficulty in finding venous access; All patients would be prophy-oriented once-weekly |
| 6–18 years old | 0 | 3 | 2 | 3 | 5 | The age group that often becomes less adherent to the prophylactic treatment; Lower acceptability of therapy in a particularly critical age group; Adolescents' crisis: prophylaxis rejection; All patients would be oriented to once-a-week |
| 18–30 years old | 2 | 1 | 6 | 2 | 2 | All patients would be once-weekly prophy oriented. |
| 30–65 years old | 1 | 6 | 1 | 2 | 3 | All patients would be once-weekly prophy oriented |
| >65 years old | 5 | 1 | 1 | 2 | 4 | Better compliance and because patients are less symptomatic and on which aggressive protection of the joints is not necessary; All patients would be once-weekly prophy-oriented. |

*3.10. Defining the Profile of Patients Eligible for Once-Weekly Prophylaxis*

Among the factors which help to define the ideal profile of patients eligible for continuous once-weekly prophylaxis, 6 out of 14 (42.8%) and 4 out of 14 (28.5%) HCs did not consider the weekly prophylaxis to be safe enough for competitive or non-competitive sports (score 1) or employment (score 1). Six out of 14 HCs (42.8%) judged weekly prophylaxis to be useful (score 6) for moderate bleeding phenotypes.

**4. Discussion**

This survey was designed to define the profile of the patient with hemophilia B eligible for once-weekly treatment with Nonacog alfa. The HTCs were first asked about the percentage distribution of their patients on replacement treatment based on the disease severity (Table 1). The number of HTCs recommending exclusively On-demand or Prophylaxis decreased or increased, respectively, according to the disease severity. The number of HTCs recommending both Prophylaxis and On-demand increased with the severity of the disease, from 1 to 11. While On-demand was the only one regimen prescribed by HTCs in the Mild hemophilia group, in the group of moderate or severe disease, the prescriptions for both Prophylaxis or On-demand, the ratio Prophylaxis/On-demand was 0.70 and 3.02, respectively.

As far as the age of patients is concerned, very few HTCs treated their hemophilia B patients exclusively by On-demand or Prophylaxis: the majority adopted both regimens. There is an inverse relationship between the age of patients and the ratio Prophylaxis/On-demand, in the group of patients treated with both regimens: the highest (4.19) and the lowest (0.35) value resulted in the younger and older patients, respectively. A continuous increase in On-demand treatment was observed with the increase of the age of patients (Table 2). This policy means a high level of care for the primary prophylaxis.

The majority of HTCs assigned the maximum score to bleed frequency as one of the clinical factors conditioning the choice of continuous prophylaxis over "On-demand" therapy regarding this factor as the primary motivation in the choice between the two modes of treatment, especially in younger age groups. Bleeding severity was also attributed to high scores in all age groups, up to 30 years in particular.

In contrast, the scores given by the HTCs to venous access were distributed relatively evenly across all age groups, suggesting this issue is not a decisive factor in the choice of the treatment strategy.

The majority of HTCs considered treatment compliance to be a significant factor, particularly in the 6–18, 18–30, and 30–65 age groups, while in the <6 and >65 year groups, the opinion of the HTCs was less homogeneous.

The majority of the HTCs attributed little importance to pro-thrombotic comorbidities in the age groups up to 18–30 years, these being at the lowest risk, while most of the respondents gave the maximum score to pro-hemorrhagic comorbidities in patients aged >65 years, comprehensible given the complex clinical picture (cardiac arrhythmias, valvular disease, etc.) of the patients in this age group.

Concerning laboratory parameters, the majority of the HTCs believe that baseline FIX at diagnosis, in combination with the clinical picture, plays a significant role in the choice between continuous prophylaxis and "On-demand" therapy in all age groups, baseline FIX being the predominant factor in the choice of the treatment regimen.

It should also be noted that knowledge of the particular mutation present in patients with hemophilia B may allow prediction of the risk of developing antibodies against infused FIX, especially about the likelihood of severe allergic and anaphylactic reactions, which have been demonstrated to occur in at least 26% of patients with a complete deletion of the *F9* gene [29,30]. When the importance of identifying the type of genetic mutation is evaluated, the majority of the HTCs expressed a medium-high score in the younger age groups and a low score in the patients aged >65 years.

Concerning socio-cultural barriers and quality of life, the majority of the HTCs gave a medium-high score to these issues in all age groups, attributing these parameters a reasonably significant role in the choice of continuous prophylaxis over "On-demand" therapy. It was noted that the use of constant prophylaxis in childhood (as well as in adolescence and adulthood) might improve quality of life: not only by reducing the need for medical appointments, use of emergency services, first aid, hospital admissions, and surgical and rehabilitative procedures, but also because it enables patients to lead a healthy life, play sports, and acquire adequate training to allow regular inclusion in social life and employment [31,32].

Most of the HTCs reported using clinical parameters as part of the regular monitoring of patients on continuous prophylaxis. Some HTCs (32%) reported that they regularly monitor patients on continuous prophylaxis by assessing the pre-infusion level of residual FIX, while a minority of HTCs (9%) reported performing pharmacokinetic studies of the FIX infusion. The survey results show conflicting opinions regarding FIX level measurements as part of regular monitoring during prophylaxis. Many experts believe that consistently maintaining levels at >1% does not in itself represent a target of prophylaxis since any dose or administration frequency adjustments are made based on clinical criteria [33]. While performing pharmacokinetic studies is undoubtedly useful, consideration must be given to its real clinical utility, taking into account the age of the patient, ease of venous access, and the need to take numerous blood samples for this type of study [34]. In any case, the pharmacokinetics should be determined following the criteria established by the Scientific and Standardisation Committee of the International Society on Thrombosis and Haemostasis [35], using the standard of the specific product being infused into the patient as the reference plasma [36].

About the administration method of continuous prophylaxis, treatment is given twice weekly to the majority of patients with hemophilia, while only 50% of HCs report treating 10–30% of their HB patients with a once-weekly prophy. As regards the dose of FIX concentrate used, 93% of the HTCs reported using a dosage of between 25–50 IU in patients on twice-weekly prophylaxis. This data

agrees with that reported in the AICE recommendations [37], showing that primary prophylaxis generally involves the administration of 40 IU/kg of FIX concentrate on two non-consecutive days per week [38]. Monitoring individual clinical responses and measuring circulating FIX levels at least 72 h after the previous infusion (trough) are recommended. This allows personalization of the dose to maintain trough FIX levels above 1–2 IU/dL [33]. In the context of a plan to use early primary prophylaxis, almost all the HTCs believed that once-weekly treatment with Nonacog alpha might represent an alternative strategy to dose escalation.

Limited data is available on the optimal prophylactic FIX replacement regimen for patients with hemophilia B [24]. Nevertheless, the safety and efficacy of Nonacog alfa have been demonstrated across a range of patient populations, including previously untreated and treated adults and children in "On-demand," preventive, and prophylactic settings [23,24,39]. In an international, multicenter, open-label, single-cohort study, Shapiro et al. demonstrated that Nonacog alfa is safe and clinically effective in the treatment and prevention of bleeding in previously untreated patients with severe or moderately severe hemophilia B [39].

In a multicenter, randomized, open-label, four-period, crossover trial conducted in previously treated patients with severe or moderately severe hemophilia B, Valentino et al. demonstrated that secondary prophylaxis therapy with Nonacog alfa 100 IU/kg once-weekly may be a safe and effective alternative to twice-weekly prophylaxis at 50 IU/kg and that both regimens reduced ABR relative to "On-demand" dosing [23]. In a multicenter, open-label trial, Kavakli et al. studied the use of once-weekly prophylaxis with Nonacog alfa 100 IU/kg in patients with moderately severe to severe hemophilia B [24], and their data supported findings from a previous study by Valentino et al. suggesting that a 100 IU/kg once-weekly prophylaxis regimen may be a viable option for patients with hemophilia B [23].

In early primary prophylaxis, some HTCs (14%) do not consider once-weekly treatment with Nonacog alfa to be an attractive therapeutic approach where dose escalation is concerned, due to the half-life of the concentrate. The half-life of FIX activity following Nonacog alfa administration has been reported to be 22–24 h in patients aged 12–61 years receiving 75 IU/kg doses who were sampled for 72 h [21], but there is some pharmacokinetic evidence to support the efficacy of once-weekly prophylaxis. FIX activity following Nonacog alfa administration persists longer than previously thought [2]. Although the critical threshold of FIX activity required for prophylaxis is not known, moderate, and mild phenotypes are associated with fewer bleeding events [3]: in this setting, clinicians use estimates of half-life to design prophylactic regimens [20]. Some of the HTCs surveyed did not consider once-weekly treatment with Nonacog alfa to be an alternative strategy to dose escalation, deeming twice-weekly prophylaxis therapy to be more appropriate, following the AICE recommendations [37]. Additionally, given the heterogeneity of the clinical phenotype, the amount of drug used in once-weekly prophylaxis may not be sufficiently well-defined to tailor treatment accurately.

In terms of the advantages of using a once-weekly infusion scheme in patients on continuous prophylactic treatment, the distribution of the answers given by the HTCs was heterogeneous, suggesting that none of the factors listed in the questionnaire seem to be a decisive factor when opting for this treatment regimen.

Similar heterogeneity was observed concerning obstacles to the use of a once-weekly prophylactic regimen in patients with difficult venous access.

These findings agree with those reported in the literature. The potential benefits of prophylactic treatment, in addition to preventing joint damage and reducing bleeding events, include less time off work, fewer hospital admissions, less frequent monitoring, and improved quality of life [40,41]. Once-weekly prophylaxis may be a viable therapeutic choice for patients with hemophilia B, offering several advantages including decreased frequency of infusions, which may be a favorable option for individuals in whom venous access is a concern, potentially improving adherence and being more convenient for patients and their caregivers, compared with more frequent prophylactic dosing regimens [23,24]. Although prophylaxis is considered an optimal approach for the treatment of severe hemophilia B [7,9], it is still underused [10,11]. Factors implicated in this underuse include difficulty

in venous access, particularly with primary prophylaxis in children, as well as overall adherence and willingness to commit to a demanding treatment schedule [41,42].

The frequency of infusions required for prophylaxis may be perceived as an obstacle to this treatment in young children. As prophylactic regimens for the treatment of hemophilia B have historically been dosed twice weekly, the once-weekly regimen provides a viable treatment option, with less frequent infusions [24]. With weekly prophylaxis, young patients can potentially gain the crucial benefits of prophylaxis, namely modest infusion schedules and no need for adjuvant devices [43].

Almost half the HTCs (46%) considered that once-weekly prophylaxis might be necessary for the >65 years age group, while another 46% did not think this regimen would be beneficial in this setting.

Clinical studies on the safety and efficacy of prophylactic regimens with Nonacog alfa have not included a sufficient number of subjects aged ≥65 years to be able to determine whether such subjects respond differently from younger subjects [44]. In these cases, the best approach to choosing the optimal treatment is to create a personalized "made to measure" prophylactic regimen (tailored prophylaxis). Such an approach requires an accurate overall clinical assessment and careful monitoring, taking the patient's needs, requirements, and lifestyle into account [24].

One of the factors that contribute to defining the ideal profile of the patient eligible for continuous once-weekly prophylactic treatment is a mild bleeding phenotype, and the majority of HTCs assigned the maximum score to this factor. While individuals with mild hemophilia B do not suffer from spontaneous bleeding, abnormal bleeding occurs with surgery, tooth extractions, and significant injuries if the condition is left untreated. Frequency of bleeding may vary from once a year to once every ten years, and patients with mild hemophilia B are often not diagnosed until later in life when they undergo surgery or tooth extraction or suffer serious injury [45]. A careful evaluation of the patient's bleeding phenotype is critical to initiating therapy before the onset of the mechanisms of synovitis or cartilage damage, which quickly lead to the beginning of hemophilic arthropathy [25]. This condition is the most common cause of morbidity in patients with hemophilia, has a significant impact on the quality of life, and generally becomes pronounced at an early age (15–25 years of age) [46].

In this regard, functional musculoskeletal status (Haemophilia Joint Health Score = 0) was considered by the majority of the HTCs as one of the most critical factors in defining the ideal profile of the patient eligible for once-weekly prophylactic treatment. Manco-Johnson et al. indicated the importance of beginning prophylactic treatment before a patient suffers from joint bleeds: once joint bleeds occur, any resulting pathological changes are irreversible, even at a very early age [47], and prophylaxis is the only treatment able to prevent or delay joint impairment [48].

Another characteristic of patients eligible for once-weekly prophylaxis is difficult venous access since the decreased frequency of infusions provides an ideal treatment option for preserving venous access [24]. Of the factors which define the perfect profile of the patient eligible for continuous prophylactic treatment using once-weekly infusions, the majority of the HTCs considered improved patient compliance to be of great importance. Since hemophilia B is a chronic condition necessitating long-term treatment, patients tend to have poor acceptance of the illness, sometimes refusing to comply with the treatment protocol. Once-weekly prophylaxis with less frequent infusions improved adherence [24]: also, the clinical benefits of this prophylactic regimen enhanced patients' quality of life, keeping them healthy and allowing them to live normal lives, participate in physical activities, and remain in employment [31,32,47].

Physical activity and sport may play a role in the maintenance of joint health in patients with hemophilia by improving muscle strength and proprioception, although the evidence for this is currently lacking [48]. The majority of the HTCs surveyed attributed some importance to competitive and non-competitive sports and to employment, including them among the factors that should be considered when defining the profile of patients suitable for once-weekly prophylactic treatment with Nonacog alpha.

## 5. Conclusions

In conclusion, from an analysis of the responses given by 14 Italian HTCs who participated in this survey, once-weekly prophylaxis with Nonacog alfa is considered an ideal treatment for patients with mild hemophilia with excellent joint and muscle status and difficult venous access. In this setting, once-weekly prophylaxis with Nonacog alfa may be the most appropriate approach, improving the cost/effectiveness ratio, promoting adherence to the prescribed regimen, and improving quality of life, due to the ability to participate in sporting activities, and remain in employment, thus allowing them to lead a healthy life.

**Author Contributions:** D.C., R.D.C., P.G., S.L., S.M., R.M., A.C.M., A.R., C.S., P.S., S.S., G.T., A.T., and E.Z. took part in the Advisory Board, filled the questionnaire, discussed the issues, reviewed, and corrected the manuscript. M.M. elaborated on the data and wrote the draft of the paper. All authors have read and agreed to the published version of the manuscript.

**Funding:** This research received no external funding.

**Acknowledgments:** All participants to the Advisory board and MM received a fee from CDM Milan—OmnicomHealthGroup.

**Conflict of Interest:** The Authors declare no conflict of interest.

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
