# Peer review of "Identification of the Profile of the Patients with Hemophilia B Eligible for Treatment with Nonacog Alfa Once-Weekly"

_reports, doi:10.3390/reports3010003_

Round 1
Reviewer 1 Report
Benefix has been around for a very long time. How it is being used in Italy is of interest especially using it once or twice daily.
The manuscript is much too long, hard to follow and too many tables.
A shorter version, focusing on the heterogeneity of clinic practices, and who benefits from 1 or 2 weekly dosing and impediments leading to the final therapeutic choices would be more concise, easier to follow, and an addition to the literature.
Comments:
PG – 2 – L92 – The threat of transfusion transmitted disease is NO LOGER and issue from plasma derives factor IX.
PG – 15 L 303-305 – How does the genetics information enable better management except at inception of the therapy and concerns for anaphylaxis.
Author Response
PG – 2 – L92 – The threat of transfusion transmitted disease is NO LOGER and issue from plasma derives factor IX.
Response: I gladly accepted to remove on page 2 – L92 the topic of viral safety of plasma derivatives, now limited to viruses of modest importance, such as B19 Parvovirus or TTV, very resistant to viricidal methods.
PG – 15 L 303-305 – How does the genetics information enable better management except at inception of the therapy and concerns for anaphylaxis.
Response: I also removed on page 15 L 303-305 the issue of the relationship between FIX genotype / FIX PK, a very complicated topic that deserves more exposure.
I would like to express my thanks to you for the advice you have sent me which have made it possible to improve the quality of my manuscript.
I reduced the number of tables to 4 and the length of the text to 5062 words. For your convenience and approval, all changes to the text are tracked in red in the new manuscript. I hope these changes will make my manuscript more acceptable
Reviewer 2 Report
Dear author. Your work is important for hematologists. I am a pharmacologist and clinical pharmacologist. I am pleased to see a deep analysis on a small number of patients. Good luck with further research.
Author Response
Dear Reviewer,
I would like to thank you for the kind words of appreciation about my manuscript. In my opinion, it is important during these times of increasing cost of hemophilia therapy, to have a look also to the cost/effectiveness ratio.